# Bayesian nowcasting with leading indicators applied to COVID-19 fatalities in Sweden

**Fanny Bergström** *, **Felix Günther, Michael Höhle, Tom Britton**

Department of Mathematics, Stockholm University, Stockholm, Sweden

* fanny.bergstrom@math.su.se

## Abstract

The real-time analysis of infectious disease surveillance data is essential in obtaining situational awareness about the current dynamics of a major public health event such as the COVID-19 pandemic. This analysis of e.g., time-series of reported cases or fatalities is complicated by reporting delays that lead to under-reporting of the complete number of events for the most recent time points. This can lead to misconceptions by the interpreter, for instance the media or the public, as was the case with the time-series of reported fatalities during the COVID-19 pandemic in Sweden. Nowcasting methods provide real-time estimates of the complete number of events using the incomplete time-series of currently reported events and information about the reporting delays from the past. In this paper we propose a novel Bayesian nowcasting approach applied to COVID-19-related fatalities in Sweden. We incorporate additional information in the form of time-series of number of reported cases and ICU admissions as leading signals. We demonstrate with a retrospective evaluation that the inclusion of ICU admissions as a leading signal improved the nowcasting performance of case fatalities for COVID-19 in Sweden compared to existing methods.

## Author summary

Nowcasting methods are an essential tool to provide situational awareness in a pandemic. The methods aim to provide real-time estimates of the complete number of events using the incomplete time-series of currently reported events and the information about the reporting delays from the past. In this paper, we propose a Bayesian approach applied to COVID-19 fatalities in Sweden. We incorporate regression components into the Bayesian hierarchical model to accommodate additional information provided by leading indicators such as time-series of the number of reported cases and ICU admissions. We use a retrospective evaluation covering the second (alpha) and third (delta) wave of COVID-19 in Sweden to assess the performance of the proposed method. We demonstrate that the inclusion of ICU admissions as a regression component improved the nowcasting performance (measured by the CRPS score) of case fatalities for COVID-19 in Sweden by 3.9% compared to when this information was not incorporated into the model.

**Data Availability Statement:** The COVID-19 surveillance data used for the analysis and R-code is openly-available from https://github.com/fannybergstrom/nowcasting_covid19.

**Funding:** TB is grateful for financial support from NordForsk, grant no. 105572 (https://www.nordforsk.org/). The computations and data handling was enabled by resources provided by the Swedish National Infrastructure for Computing (SNIC) at HPC2N partially funded by the Swedish Research Council through grant agreement no. 2018-05973 (https://www.snic.se/). The funders had no role in study design, data collection and analysis, decision to publish, or preparation of the manuscript.

**Competing interests:** The authors have declared that no competing interests exist.

## Introduction

The real-time analysis of infectious disease surveillance data is one of the essential components in shaping the response during infectious disease outbreaks such as major food-borne outbreaks or the COVID-19 pandemic. Public health agencies and governments typically monitor disease dynamics using time-series of reported cases or fatalities to assess the effectiveness of preventive measures and plan further actions [1, 2]. Such real-time analysis is complicated by reporting delays that give rise to *occurred-but-not-yet-reported* events which may lead to underestimation of the actual number of events. Fig 1 illustrates the problem with data of Swedish COVID-19-related fatalities as of 2022–02-01. While the reported number of fatalities per day suggested a declining trend, data available two months later [3] revealed that the number at the time was actually increasing.

Nowcasting methods [4–6] tackle this problem by providing real-time estimates of the complete number of events using the incomplete time-series of currently observed events and information about the reporting delay from the past. The methods have connections to insurance claims-reserving [7] and its epidemiological applications trace back to HIV modelling [8–10]. Nowcasting methods have been used in COVID-19 analysis for daily infections [11–13] and fatalities [14–16]. The foundation of our method is a Bayesian approach to nowcasting and was initially developed by Höhle and an der Heiden [5] and later extended by Günther et al. [17] and McGough et al. [6].

Most nowcasting methods are focused on estimating the reporting delay distribution. However, an epidemic contains a temporal dependence and adheres to certain "laws", for instance slow changes in contact behavior. Furthermore, with air-born diseases such as COVID-19, the existing number of infectees will influence the number of future infections. Taking this temporal dependence of the underlying disease transmission into account has been shown to

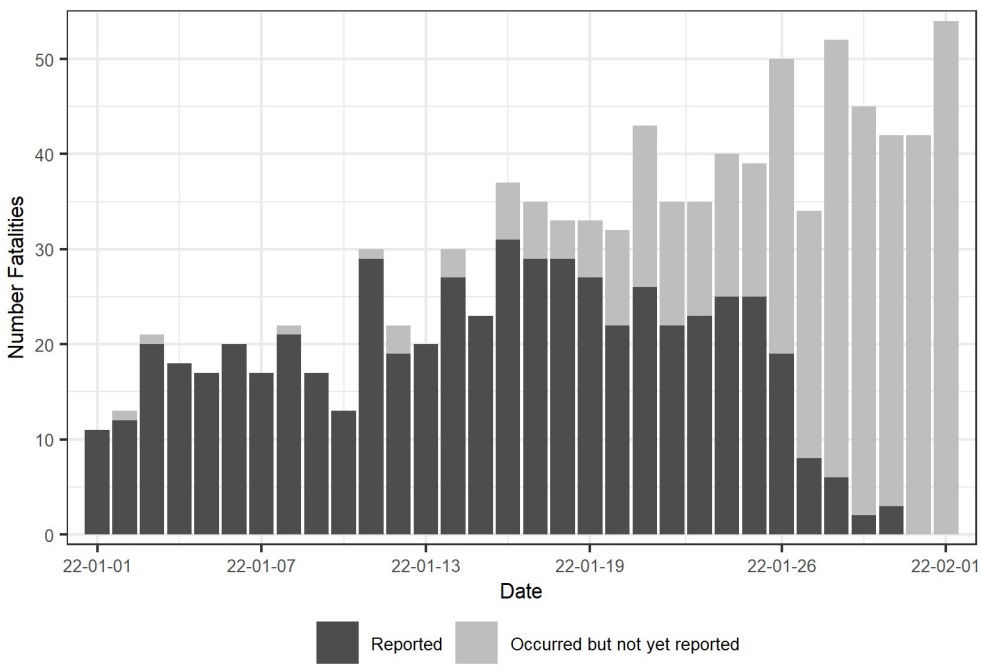

**Fig 1. Daily COVID-19 fatalities in Sweden.** Reported (black bars) and unreported (grey bars) number of daily fatalities as of 2022–02-01. The reported number of events show a declining trend when in actuality (known in hindsight) it was increasing.

improve the nowcasting performance [6, 17]. Another approach to nowcasting is to use other data sources that are sufficiently correlated with the time series of interests, for example demonstrated in the Machine Learning approach by Peng et al. [18]. Bastos et al. [19] propose a generalized linear model (GLM) based approach [20] to correct for reporting delays which can account for covariates and spatial random effects, a method that Miller et al. [21] applies to nowcasting Chikungunya fever using Google searches as a covariate.

Our approach for nowcasting Swedish COVID-19 fatalities is based on a flexible Bayesian hierarchical model that can account for temporal changes in the reporting delay distribution and handle various reporting structures. As an extension to existing methods [5, 17] this method incorporates a regression component of additional correlated data streams. The disease stages (infected, hospital, ICU, death) have a time order and the number of new entries in one of the earlier compartments can help estimate what will happen in the later stages. We evaluate the time-series of the number of Intensive Care Unit (ICU) admissions and reported cases as additional correlated data streams. We assume that these data streams will be informative of the fatalities and use these as leading indicators in our Nowcasting model.

In this paper we present the methodological details of our approach and compare the results to existing nowcasting methods to illustrate the implication of incorporating additional data streams associated with the number of fatalities. We demonstrate with a retrospective evaluation of our method that nowcasting with leading indicators can improve the predictive performance compared to existing methods.

## Materials and methods

### Data

The surveillance data used for the analysis in this paper are daily counts of fatalities, ICU admissions and reported cases of people with a laboratory-confirmed SARS-CoV-2 infection in Sweden. The period ranges from 2020–10-20 to 2021–05-21 and contains 117 reporting days (Tuesday to Friday excluding public holidays). During this period, there were 951 646 reported cases, 4 734 ICU admissions and 8 656 fatalities. The evaluation period covers Sweden's second (alpha) and third wave (delta) of COVID-19-related fatalities. In addition, this period also covers the introduction of vaccination which meant a change in the association between reported cases or ICU admissions and the fatalities. The times series of the number of reported cases, ICU admissions and deaths can be seen in Fig 2. The figure shows that the rise and fall of the three time series follow a similar time trend. During the first wave the rise and fall of the three time series follow a similar time trend with a time shift as the earlier disease compartments are ahead in time. In the second wave the relative association between the fatalities and the other disease stages becomes less substantial, the main reason being the introduction of the nationwide COVID-19 vaccination program that started 2020–12-27.

The data used in our analysis is publicly available from the website of the Public Health Agency of Sweden [3], where new reports have been published daily from Tuesday to Friday (excluding public holidays). The aggregated daily counts are updated retrospectively at each reporting date. As the case fatalities are associated with a reporting delay, the published time series of reported COVID-19 fatalities will always show a declining trend (see Fig 1 for an illustrative example). The reporting delay can not be observed in a single published report but can be obtained by comparing the aggregated numbers of fatalities of each date from previously published reports.

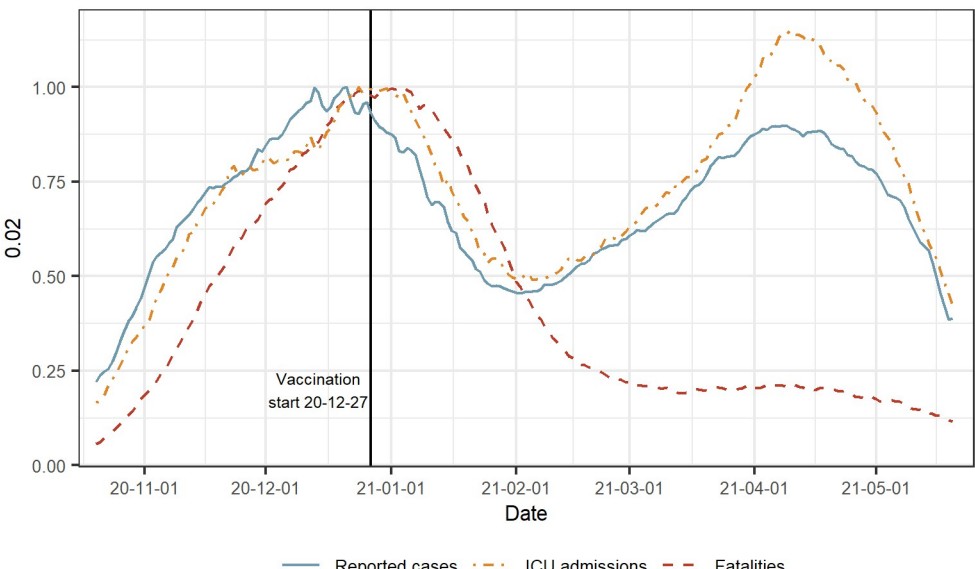

**Fig 2. Reported cases, ICU admissions and fatalities with COVID-19 in Sweden.** The period covers the second (alpha) and third (delta) wave and the start of vaccination in Dec 2020. Each time series is shown with a 3-week centered rolling average and scaled by its maximum value in the peak around Dec 2020.

## Nowcasting

The notation and methodological details of our approach follows closely the notation introduced in Günther et al. [17]. Let $n_{t,d}$ be the number of fatalities occurring on day $t = 0, \ldots, T$ and reported with a delay of $d = 0, 1, 2, \ldots$ days, such that the reporting occurs on day $t + d$. The goal of Nowcasting is to infer the total number of fatalities $N_t$ of day $t$ based on the information available on the current day $T \geq t$. The sum $N_t$ can be written as

$$N_t = \sum_{d=0}^{\infty} n_{t,d} = \sum_{d=0}^{T-t} n_{t,d} + \sum_{d=T-t+1}^{\infty} n_{t,d}, \tag{1}$$

where the first sum is observed and the second sum is yet unknown. This can be illustrated by the so called reporting triangle (Fig 3). Where the upper left triangle are the number of reported fatalities and the lower right triangle is the number of occurred- but-not-yet-reported events with a maximum delay of $D$ days. The upper triangle carries the information about the reporting delay from the past and the lower triangle is what is estimated with the Nowcasting model.

We let $\lambda_t$ denote the expected value of $N_t$, and $p_{t,d}$ denote the conditional probability of a fatality occurring on day $t$ being reported with a delay of $d$ days. Then, the number of events occurring on day $t$ with a delay of $d$ days is assumed to be negative binomial distributed

$$n_{t,d}|\lambda_t, p_{t,d} \sim \mathrm{NB}(\lambda_t \cdot p_{t,d}, \phi),$$

with mean $\lambda_t \cdot p_{t,d}$ and overdispersion parameter $\phi$. Hence, the Nowcasting task can be seen as having two parts; (1) determine the expected value of the total number of fatalities and (2) determine the reporting delay distribution to subsequently predict the $n_{t,d}$'s and finally compute the $N_t$'s.

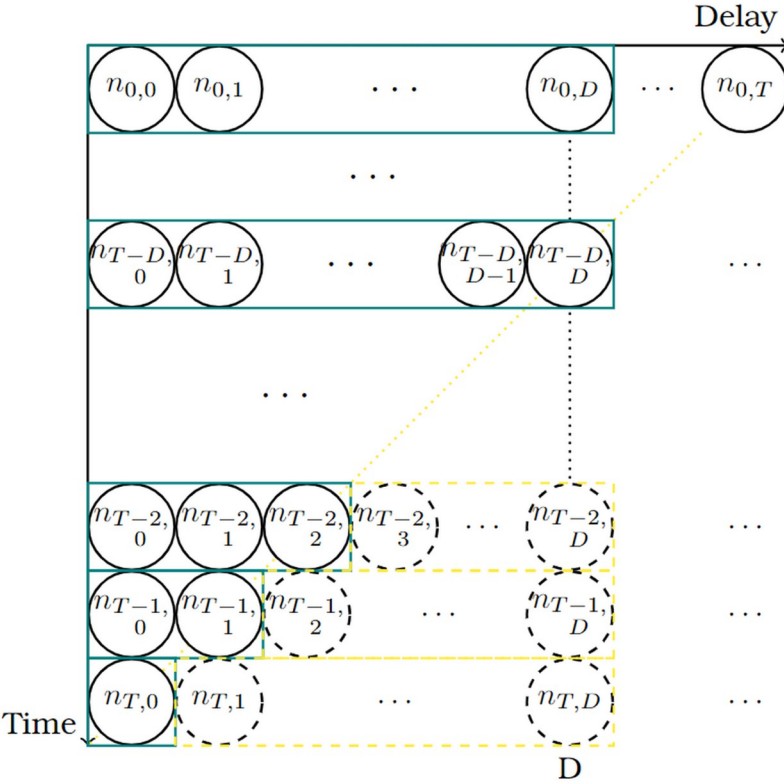

**Fig 3. Reporting triangle for day $T$.** Green boxes (solid line) where $t \leq T - D$ are the reported number of fatalities on day $T$ (today) with a maximum delay of $D$ days. The yellow boxes (dashed line), corresponding to $t > T - D$, are the occurred-but- not-yet-reported number of events of day $t + D$.

## Flexible Bayesian nowcasting

As described in the previous section the nowcasting problem can be seen as a problem of the joint estimation of two models: (1) a model for the expected number of deaths over time, and (2) a model for the reporting delay distribution. Therefore, we let our model constitute of two distinct elements; (1) the underlying epidemic curve determining the expected number of fatalities $\lambda_t$ and (2) the reporting delay distribution determining $p_{t,d}$. We will in the following describe the structure of each.

**Component 1: The expected number of fatalities.** Let $\lambda_t = \mathbb{E}[N_t]$ denote the expected total number of fatalities occurring on day $t$. We specify a baseline model for $\lambda_t$ as

$$\log(\lambda_t)|\lambda_{t-1} \sim N(\log(\lambda_{t-1}), \sigma^2), \tag{1}$$

where $t = 0, \ldots, T$ and $d = 0, \ldots, D$. Time $t = 0$ is assumed to be the start of the observation period, such as the start of the pandemic or a new wave. This approach to model $\lambda_t$ as a random walk on the log scale is proposed by McGough et al. [6] and Günther et al. [17]. Here we will refer to it as model R.

An alternative to model R in Eq (1) is to assume that we can predict the total number of fatalities with additional data streams associated with the event of interest. The additional data streams are assumed to be ahead in time compared to the time series of interest, for example due to the tracked event of the data stream being at an earlier stage in a typical COVID-19 disease progression or because of a smaller reporting delay. Therefore we may use the additional

data stream as a leading indicator in the Nowcasting model. One approach is to consider the number of fatalities as some time-varying fraction of the numbers in the additional data streams. Let $M_t = (m_{1,t}, \ldots, m_{k,t})$ denote a vector of $k$ leading indicators at time $t$. We specify a regression type model for $\lambda_t$ as follows

$$\log(\lambda_t)|M_t \sim N(\beta_0 + \beta'M_t, \sigma^2), \tag{2}$$

where the $\beta_0$ is an intercept and $\beta$ denotes the vector of additive effects of the $k$ data streams on the log of the mean of $\lambda$. With this model specification we assume a strong association between the case fatalities and the $k$ data streams measured some days earlier. We will refer to this model as L($M$).

Furthermore, we propose another approach combining the random walk component of the model in Eq (1) and the additional data streams of Eq (2). We let the leading indicators be the change in the additional data streams such as case reports or hospitalizations. In other words we assume that if there is an increase in the leading indicator, we also expect an increase in the number of fatalities. An increase in an earlier disease compartment as case reports is not expected to give an instant increase in the number of deaths but rather with some time delay, so as for the model in Eq (2), the leading indicators need to be specified with a suitable time delay. We specify this alternative model for $\lambda_t$ as

$$\log(\lambda_t)|\lambda_{t-1}, M_t \sim N(\log(\lambda_{t-1}) + \beta'M_t, \sigma^2), \tag{3}$$

where $\beta$ is again the vector of regression coefficients for the $k$ leading indicators $M_t$. This approach combines an established method [17] with additional information that is informative of the events of interest. We note that when the $\beta$-coefficients of this model are zero, this model becomes identical to the model specified in Eq (1). This model will be referred to as RL ($M$). In related pre-pandemic work, Bastos et al. [19] propose a hierarchical Gaussian Markov Random Field and GLM approach in order to handle nowcasting in setting with covariates. A theoretical treatment of the differences between our model and their approach is provided in S1 Appendix Sec 7.

**Component 2: The reporting delay distribution.** The model for the reporting delay distribution at day $t$ is specifying the probability of a reporting delay of $d$ days for a fatality occurring on day $t$. We denote this conditional probability

$$p_{t,d} = P(\text{delay} = d|\text{fatality day} = t).$$

Similarly to Günther et al. [17], we model the delay distribution as a discrete time hazard model $h_{t,d} = P(\text{delay} = d|\text{delay} \geq d, W_{t,d})$ as

$$\text{logit}(h_{t,d}) = (\gamma_d + W'_{t,d}\eta) \times Z_{t,d}, \tag{4}$$

where $d = 0, \ldots, D-1$, $h_{t,D} = 1$, $\gamma_d$ is a constant, $W_{t,d}$ being a vector of time- and delay-specific covariates and $\eta$ the covariate effects. The distinction from Günther et al. [17] is the $t \times d$ matrix $Z$ which is an indicator for non-reporting days. The matrix has elements $Z_{t,d}$ that takes values 1 when day $t + d$ is a reporting day and 0 otherwise. It can be shown how the reporting probabilities are derived from Eq (4) [17]. We are using linear effects of the time on the logit-scale with break-points every two weeks before the current day to allow for changing dynamics in the reporting delay distribution over time. We also use a categorical weekday effect to account for the weekly structure of the reporting.

## Inference and implementation

Inference for the hierarchical Bayesian nowcasting model is done by Markov Chain Monte Carlo using R-Stan [22] extending the work of Günther et al. [17]. The prior distributions used are found in S1 Appendix Sec 1. In order to ensure reproducibility and transparency, the R-Code [23] and data used for the analysis is available from https://github.com/fannybergstrom/nowcasting_covid19.

## Evaluation metrics

As in Günther et al. [17], we use the following four metrics to quantify the model performance; (1) continuous rank probability score (CRPS), (2) log scoring rule (logS), (3) root mean squared error (RMSE), and (4) the prediction interval (PI) coverage. The CRPS and logS are *proper scoring rules* that assess the quality of the probabilistic forecast using the posterior predictive distribution of the probabilistic forecast [24]. Proper scoring rules assign numerical scores to pairs of forecasts and observations and can be used to assess accuracy and sharpness of the forecast simultaneously.

Following the notation of Czado et al. [20], we let $X$ be a integer-valued non-negative stochastic variable with a realisation $x$. The nowcasts produce a probabilistic forecast quantified by the infinite vector $P$ such that $\mathbb{P}(X \leq i) = P_i$, for $i = 0, 1, 2, \ldots$. We define a vector $p$ with elements $\mathbb{P}(X = i) = p_i$ for $i = 0, 1, 2, \ldots$. We let $\hat{x}^{(P)}$ denote a point estimate for $X$ based on $P$. We also let $q_z^{(P)}$ denote the $z$ quantile of $P$, with $0 \leq z \leq 1$.

The CRPS is defined

$$\text{CRPS}(P, x) = \sum_{i=0}^{\infty} (P_i - \mathbb{1}(x \leq i))^2,$$

where $\mathbb{1}(\cdot)$ is the indicator function. The CRPS is a generalisation of the mean absolute error (MAE) for a distribution, i.e. if $P$ is a point estimate then the CRPS reduces to the MAE of the point estimate. The CRPS is negatively oriented, meaning that smaller scores indicate better predictive performance.

The logS is the negative logarithm of the predictive probability mass function evaluated at the realisation $x$. The logS is defined

$$\text{logS}(P, x) = \begin{cases} -\log p_x & \text{if } p_x > 0 \\ 0 & \text{if } p_x = 0. \end{cases}$$

Also for this score a smaller value indicates a better performance.

The RMSE assess the deterministic predictive accuracy of the point estimate $\hat{x}^{(P)}$. It is calculated as

$$\text{RMSE}(P, x) = \sqrt{(x - \hat{x}^{(P)})^2}.$$

In our application we let $\hat{x}^{(P)}$ be the median of $X$ based on $P$.

The fourth evaluation metric, the PI coverage, is used to quantify the model uncertainty. This metric indicates if the realisation $x$ is contained within the $100 \cdot (1 - \alpha)\%$ equal-tailed PI given by $P$. The PI coverage can mathematically be expressed as

$$\text{cov}_\alpha(P, x) = \mathbb{1}(q_{\alpha/2}^{(P)} \leq x \leq q_{1-\alpha/2}^{(P)}),$$

meaning that it is equal to 1 if $x$ is contained in the PI and 0 if else. We note that the PI

coverage is not a proper scoring rule since it does not entail information about the quality of the forecast beyond if the realisation is contained within the chosen PI. If the model uncertainty is well calibrated, we expect the average PI coverage over a set of time points to be equal to $1 - \alpha$.

In our application the nowcasts for one time instance $T$ produce probabilistic forecasts for $N_T, \ldots, N_{T-D}$, where $T$ is the most recent date for which new data is available and $D$ is the assumed maximum number of days reporting delay. We evaluate the estimates $\hat{N}_t$, $t = T, \ldots,$ $T - D$ for each of the $n$ time points $T$ in the evaluation period. We let $s_{t,d}$ denote the score of the evaluation of $\hat{N}_{t-d}$ estimated with the information available as of day $t$, where $t$ is the reporting day and $d = 0, \ldots, d_{max}$ is the number of days since day $t$. We let $d_{max}$, $d_{max} \leq D$, be the maximum number of days since day $t$ we choose to include in the evaluation. Over a set of time points $\{0, \ldots, n\}$, we let the mean score $d$ days since day $t$ be defined as

$$S_d = \frac{1}{n} \sum_{t=0}^{n} s_{t,d}. \tag{5}$$

We expect $S_d$ to be a decreasing function of $d$ as there will generally by less uncertainty about $N_t$ as $d$ increases which will make the nowcasting task easier. Next we define $S_t$ as the average score for the nowcasts estimated with the information available as of day $t$. We let

$$S_t = \frac{1}{d_{max}} \sum_{d=0}^{d_{max}} s_{t,d}. \tag{6}$$

Finally we define the the mean overall score $S$ as the average performance over all time points and the $d_{max}$ days since day $T$. We define $S$ as

$$S = \frac{1}{n \times d_{max}} \sum_{t=0}^{n} \sum_{d=0}^{d_{max}} s_{t,d}. \tag{7}$$

In our retrospective evaluation of the nowcasting performance we are most interested in the latest predictions as these are the most informative of the current trend of the pandemic. We therefore choose $d_{\max} = 6$ such that we evaluate the forecasts of the latest week from the reporting day $T$; $\hat{N}_T, \ldots, \hat{N}_{T-6}$ for the $n$ reporting dates $T$ in the evaluation period.

## Results

### Application to fatalities

We apply the nowcasting methods to reported COVID-19 fatalities in Sweden and let the number of reported cases and COVID-19 associated ICU admissions act as two leading indicators. The reporting of ICU admissions is also associated with a reporting delay but considerably shorter than the fatalities. We use model R as a benchmark model and compare it to the two alternative models using leading indicators. For the leading indicator time series we use a seven day centered rolling average to avoid the weekday effect of the reporting. For model L we let the leading indicator be the number of COVID-19-related ICU admissions and for model RL the leading indicator is the change in ICU admissions of two consecutive weeks. We denote the leading indicator models as L(ICU) and RL(ICU). The pre-specified lag between the fatalities and leading indicators is determined by fitting a linear time series model given the two model specifications of models L and RL and choosing the lag providing the best fit. The period chosen for the time series model is 2020–04-01–2020–10-19 to use the information available only prior to the evaluation period. We use 18 days lag for the reported cases and 14

days lag for the ICU admissions. In practice, ICU admissions are also reported with a small delay but here only 3.4% of the ICU admissions are reported with a delay above the chosen lag of 14 days, adjustments for this second reporting delay appear negligible for our application (but see also Sec Discussion). For practical and robustness reasons, we use a maximum reporting delay of $D$ = 35 days for the fatalities. For the fatalities reported with a delay longer than the maximum, we set their delay to the upper limit of 35 days. Of the case fatalities 1.3% were reported with a delay longer than 35 days during the evaluation period.

The reporting triangle for our application will have diagonal lines of cells of no reporting because of the non-reporting days (Saturday–Monday and public holidays). An illustration of the reporting triangle using reported COVID-19 fatalities in Sweden is found in S1 Appendix Sec 2. This prior knowledge about the non-reporting days is included in the reporting delay model in the following way; we explicitly set the reporting probability $p_{t,d}$ to zero for all combinations of reference $t$ and delay $d$ days where day $t + d$ is a non-reporting day. This follows directly from the $Z$-matrix and the discrete time hazard model of $h_{t,d}$ defined in Eq (4). These non-reporting days are then also excluded from the calculations of the likelihood.

## Retrospective nowcasting evaluation

A retrospective evaluation was used to assess the performance of the Nowcasting models. We use the four evaluation metrics (CRPS, logS, RMSE and PI coverage) as described in Sec Evaluation metrics. The model-based predictions are compared to the (now assumed to be known) final number of COVID-19-related reported fatalities in Sweden. The samples from the posterior predictive distribution for the estimates of the total number of reported COVID-19 fatalities for day $t$ $\hat{N}_t$, $t = T, \ldots, T - 35$ are extracted for each of the 117 reporting dates $T$ of the evaluation period. The RMSE is calculated with a point estimate being the median of the posterior predictive distribution of $\hat{N}_t$, while the scoring rules CRPS and logS takes the full posterior distribution into account. For the three numerical scores CRPS, logS and RMSE, a low score indicate a better predictive performance and for the model uncertainty to be well calibrated the PI coverage should be equal to $1 - \alpha$.

Nowcasts and the estimated reporting delay for a specific reporting date $T$ = 2020–12-30 are shown in Fig 4. In the left column, the black bars are the number of fatalities reported until day $T$ and the red dashed line is the true number, only known in retrospect. The solid lines are the median of the posterior predictive distribution of $\hat{N}_t$ and the shaded areas indicate the equal-tailed point-wise 95% Bayesian prediction interval, estimated with information available at the reporting date $T$. The right column shows the daily empirical and estimated number of days of reporting. The solid lines are the estimated and empirical median days of reporting delay and the shaded area is between the 5% and 95% quantile of the reporting delay. The lower bound indicate the number of days until 5% of the total number of fatalities will be reported and the upper bound is within how many days 95% will be reported. The empirical median and the respective quantiles are calculated with data available in hindsight and the estimated quantities are obtained with the information available at the reporting date.

We observe an underestimation of the reporting delay for the L(ICU) model for the last days in the observation window (2020–12-25–2020–12-30) resulting in an underestimation of the daily number of fatalities (Fig 4B). We can also note that the PI is more narrow for L(ICU) than for the other two models and that the true number is not always contained in the PI. Model R and RL(ICU) (Fig 4A and 4C) provide similar results with less underestimation of the reporting delay resulting in a point estimate of the median of the predictive distribution lying closer to the true number compared to model L(ICU). A difference between the performance between R and RL(ICU) is that RL(ICU) provides less wide PI than R. For R and RL

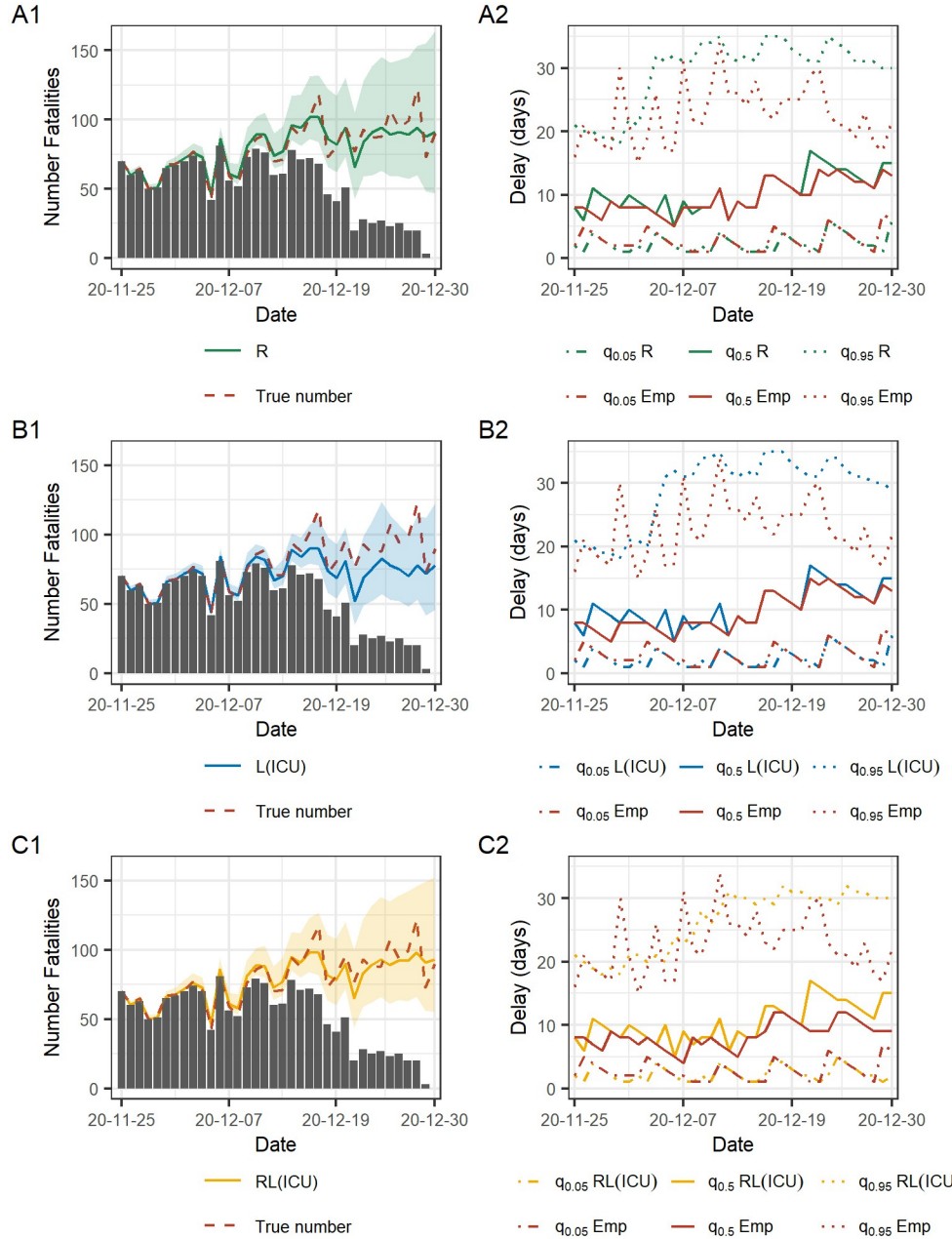

**Fig 4. Nowcasts for a specific reporting date.** Left column shows the nowcasts of 2020–12-30 where the solid lines are the median of the posterior predictive distribution of $\hat{N}$ and the shaded area depict the 95% PI. The black bars are what is yet reported and the red line is the true number, only known retrospectively. Right column shows quantiles of the estimated and empirical reporting delay distribution. The solid lines are the median reporting delay in days (for each date) and the lower and upper bounds are the 5% and 95% quantiles. The empirical quantiles are obtained with data available in hindsight.

(ICU), the true number of daily fatalities is contained in the PI for all days $T$-$t$, $t = 0, \ldots, 35$. The right column of the figure shows that the 5% quantile of the estimated number of days of reporting delay for all three models are similar to the empirical 5% quantile. Also the median of the estimated number of days reporting delay follows the corresponding empirical quantity

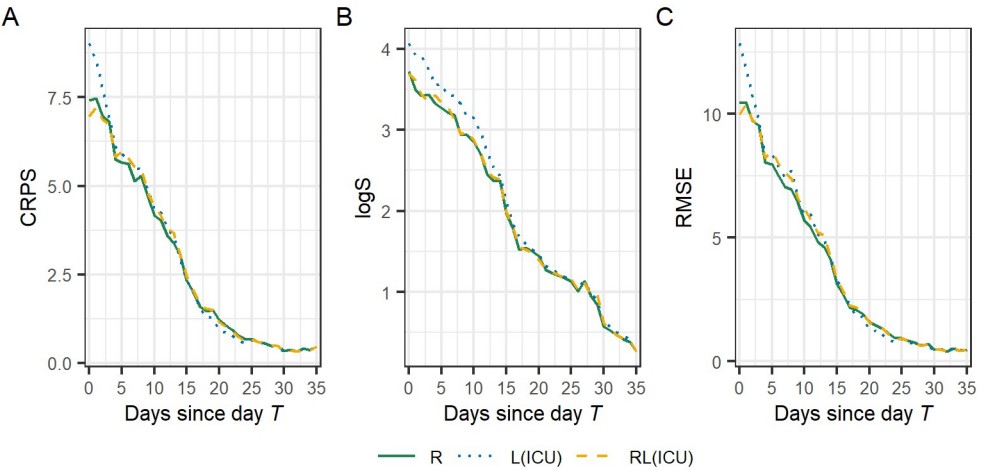

**Fig 5. Mean scores by the number of days $T$-$t$ since the day of reporting $T$.** The scores are averaged over all reporting dates $T$ in the evaluation period from 2020–10-20–2021–05-21.

reasonably well while the 95% estimated quantiles are farther from the empirical. This indicates that all three models capture the short-term trends such as the weekly reporting patterns well. On the other hand, they do not fully capture the changing dynamics of the long reporting delays given by the high spikes in the early period of the observation window and the rapid decrease in reporting delay in the final week. An alternative visualization of the empirical and estimated reporting delay distribution for the three models provided by the cumulative reporting probability is found in S1 Appendix Sec 3.1. Detailed results of the predictive performance of the nowcasting for this specific reporting date including scores, PI coverage and running times for the models are found in S1 Appendix Sec 3.2 where we also include results of using the combination of reported cases and ICU admissions as leading signals.

Seen in Fig 4, the PI is increasing in width as the final date $T$ of the observation window is approaching. As the number of days $t$ since day $T$ decreases, the uncertainty for the nowcast of day $T$-$t$ increases as the fraction of the total number of reported fatalities will be decreasing. The average score as a function of number of days $T$-$t$ as defined in Eq (5) is shown in Fig 5. For all models and scores, the score is generally a decreasing function of the number of days since day $T$. In other words, the farther from "now", the closer are the nowscast estimates of the daily number of fatalities to the true number. The most profound difference in performance for the three models is found close to day $T$ and as the number of days since day increases the model performance becomes more similar. Model RL(ICU) has a lower CRPS and RMSE score (Fig 5A and 5C) and model R has the lowest logS (Fig 5B). Model L(ICU) has the overall highest values of the scores which indicates that it has the worst performance of the three models.

The mean overall score and the coverage frequency of the 75%, 90%, and 95% prediction interval of the three models for the nowcasts performed in the evaluation period is found in Table 1. For each reporting day $T$, we use the average score of the last seven days; $T, \ldots, T-6$ as defined in Eq (7). Based on the CRPS and RMSE, model RL(ICU) has the best predictive performance, with a decrease of 3.9% and 1.0% respectively compared to model R. Model R has the lowest logS score but only with a slight advantage compared to RL(ICU) (0.38% improvement). Model L(ICU) has the worst performance for all three scores. The coverage of the prediction intervals for models R and RL(ICU) is of satisfactory levels. In contrast, the L

**Table 1. Results of the retrospective evaluation of different nowcasting models on COVID-19 related fatalities in Sweden.**

| Score | R | L(ICU) | RL(ICU) |
|---|---|---|---|
| CRPS | 6.53 | 7.04 | **6.28** |
| logS | **3.62** | 3.85 | 3.63 |
| RMSE | 9.18 | 9.95 | **9.09** |
| Cov. 75% PI | 76.92% | 66.18% | 74.97% |
| Cov. 90% PI | 91.82% | 80.95% | 89.87% |
| Cov. 95% PI | 95.85% | 88.52% | 94.99% |

CRPS is the continuous ranked probability score, logS is the log score, and RMSE denotes the root mean squared error of the posterior median. Additionally, we provide coverage frequencies of 75%, 90% and 95% credibility intervals in the estimation of the daily number of case fatalities. The scores are averaged over nowcasts for day $T, \ldots,$ $T - 6$, with $T$ being all reporting dates in the evaluation period.

(ICU) model has low coverage, indicating that the estimates of model L(ICU) is less trustworthy compared to the other models.

Fig 6 shows the retrospective true number of daily fatalities and the median of the predictive distribution of $\hat{N}_T$ and a 95% PI of the three models evaluated on each reporting day $T$ in the evaluation period. In Fig 4, this corresponds to the nowcast estimates of the final date $T$ = 2020–12-30. We observe a similar performance over time for models R and RL(ICU) (Fig 6A and 6C) and the more significant deviations from the true number appear mainly on the same reporting dates for the two models. In early Jan 2021, RL(ICU) underestimates the number of daily fatalities, likely due to the rapid decrease in ICU admissions due to the introduction of vaccines at the end of Dec 2020, while the case fatalities were also on a downwards trend but not as steep. Model RL(ICU) stabilizes after approximately two weeks (same as the length of the linear change points) in mid Jan 2021 as the model adapts to the new association between ICU admissions and case fatalities. Model L(ICU) (Fig 6B) does not have the high peaks in the posterior predictive distribution of $\hat{N}$ as the other two models. However, the deviation of the posterior median compared to the true number is visibly larger. Starting from Dec 2020, we observe an underestimation of the number of fatalities, and from Feb 2021, an overestimation for the following two months. From Apr 2021 until the end of the evaluation period, the three models have a visibly similar performance with a posterior mean close to the true number of daily fatalities and a narrow PI containing the true number. The performance of the alternative models with leading indicators compared to model R can be explained by the estimated association between the fatalities and the leading indicators. The changing dynamics of the association over time are captured by the estimated $\beta$-coefficients of the respective models. Details of the estimated $\beta$-coefficients for models R(ICU) and RL(ICU) over the evaluation period are reported in S1 Appendix Sec 4.

Looking at the predictive performance of the three nowcasting models over time, we use the seven-day average scores of the three models evaluated at the 117 reporting dates in the evaluation period as defined in Eq (6). The CRPS and logS scores are shown in Fig 7. For the three models, the scores are generally higher when the number of case fatalities is high. Overall, the performance of model R and RL(ICU) is similar, as could also be observed in Fig 6. From the beginning of the evaluation period until the end of 2020, model L(ICU) has an overall lower score and a more stable performance with less high spikes in the score compared to model R and RL(ICU). During Jan 2021, the performance is similar for the three models, but

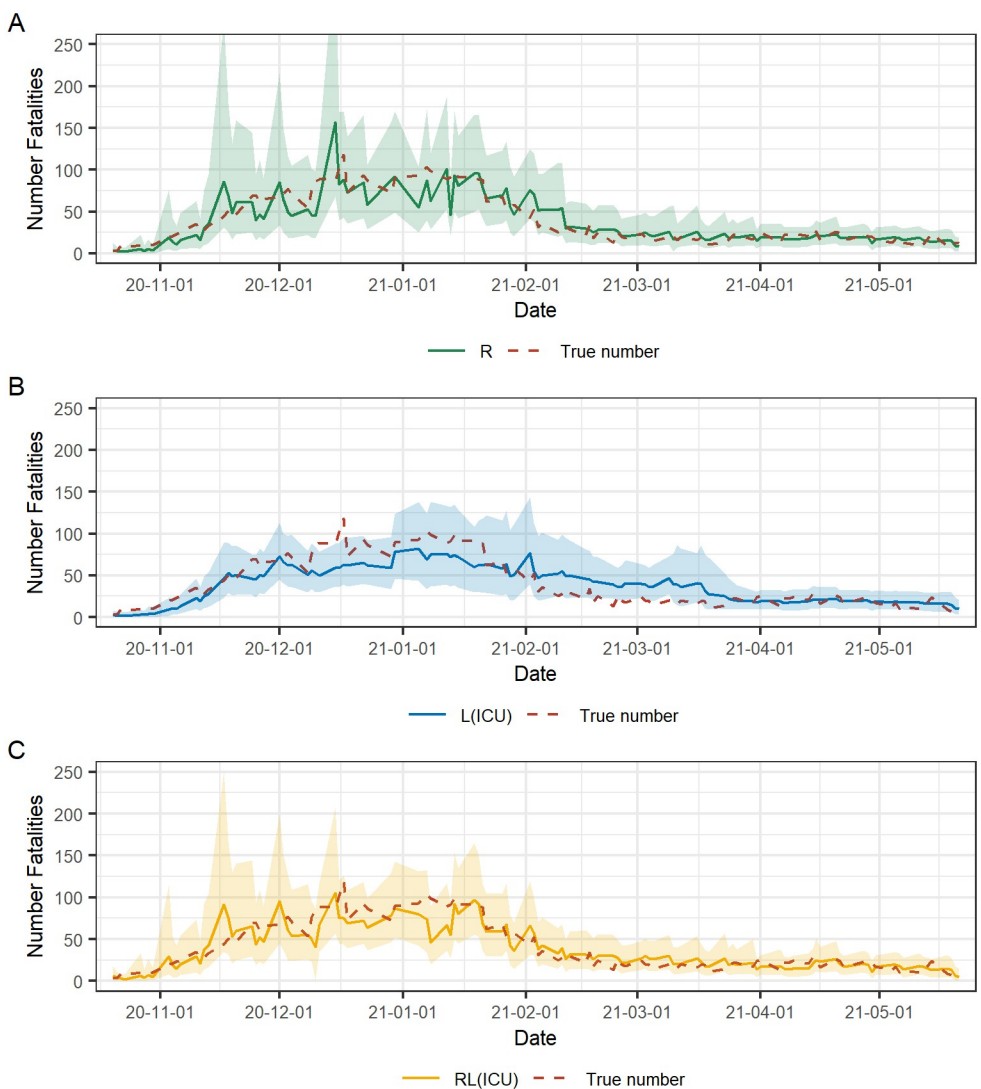

**Fig 6. Estimated and true number of fatalities with COVID-19 in Sweden.** The estimated number of fatalities are the nowcasts of day $T$ being each reporting date in the evaluation period from 2020–10–20 to 2021–05–21. The solid lines are the median of the posterior predictive distribution of the number of daily fatalities $\hat{N}_T$ and the shaded area depict the point-wise 95% PI. The red line is the retrospective true number.

from Feb to Apr 2021 model L(ICU) performs significantly worse than the other models. The remaining scoring rule, the RMSE, entails similar results (S1 Fig). After Apr 2021, the number of daily fatalities has stabilized to a low number and the score for three models becomes similar until the end of the evaluation period.

In conclusion, we find that model R and model RL(ICU) perform well over the evaluation period and has a satisfactory level of PI coverage. Furthermore, model RL(ICU) provided the best performance of the three models, indicating that there is a gain (3.9% decrease in CRPS compared to model R) of including leading indicators. Using reported cases or the combination of reported cases and ICU admissions as leading indicators does not improve performance. The results of using these leading indicators are found in S1 Appendix Sec 5.

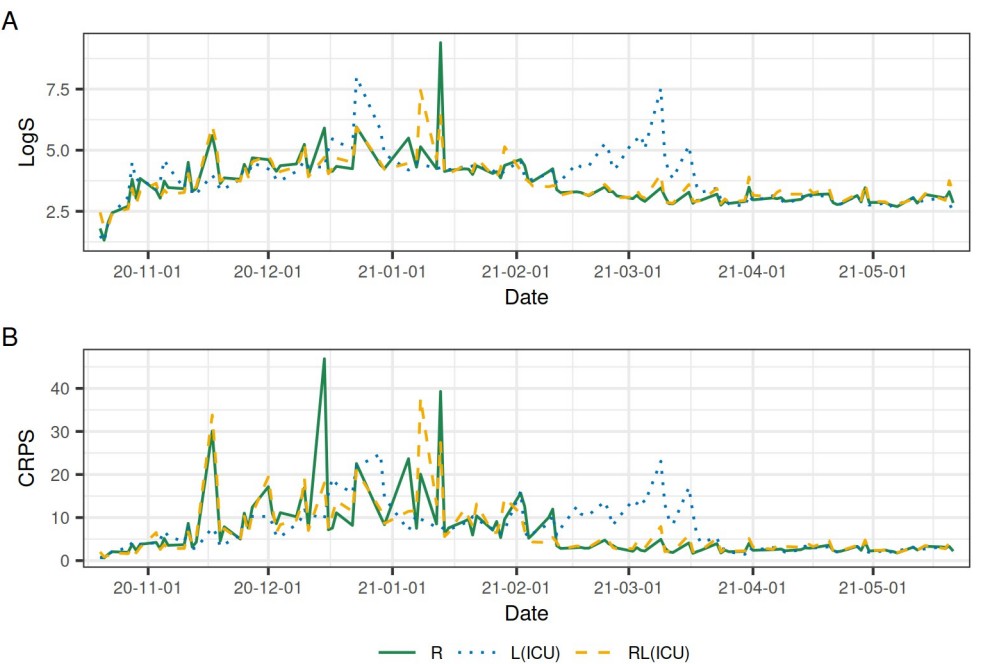

**Fig 7. Scoring rules.** Average CRPS and logS of the last 7 days; $T − 6, . . ., T − 0$ for each reporting day $T$, in the evaluation period.

## Discussion

In this paper we present an improved method for real-time estimates of infectious disease surveillance data suffering from a reporting delay. The proposed method can be applied to any disease for which the data can be put in the form of the reporting triangle given in Fig 3. We apply the method to COVID-19-related fatalities in Sweden. Even though fatalities are a lagging indicator to obtain situational awareness about the pandemic and is not without difficulties, it is often used as a more robust indicator to assess the burden of disease because it might be less influenced by the current testing strategy. Monitoring the time series of reported deaths has therefore been of importance in the still on-going COVID-19 pandemic.

We demonstrate that using leading indicators, such as the COVID-19-associated ICU admissions, can help improve the nowcasting performance of case fatalities compared to other methods. Beyond using reported cases and ICU admissions as leading indicators for the case fatalities, other possible leading indicators are vaccination, hospitalizations, and virus particles in wastewater [25], or using age-stratified reported cases. However, nowcasting with leading indicators should be made with caution and be reevaluated as the dynamics between the leading indicator and the event of interest change, which may not be a trivial task during an ongoing pandemic. Furthermore, by re-estimating the association coefficients of the leading indicator at each reporting date, our method captures the changing association between ICU admissions and case fatalities over time. However, we use a pre-specified time lag unknown at the start of the pandemic and might also change throughout the pandemic. A possible extension of our work would thus be to estimate this time lag as a part of the model fitting. Furthermore, it might also be sensible to adjust for reporting delay associated with the leading indicators. Because we use the ICU indicator as reported 14 days ago (with 96.6% of ICU cases being reported by then), the added value of such a "double nowcasting" is limited in our

application, but in settings with larger reporting delay in the leading indicators this might be different.

We use a first order random walk in model R and RL(ICU), but as a sensitivity analysis we also investigated specifying an AR(2) model for $\lambda_t$ in order to obtain more smooth nowcast estimates. Preliminary results (S1 Appendix Sec 6) showed no improved predictive performance compared to the simple random walk. Yet we do not exclude the possibility that this type of model specification could improve the model performance in other settings, e.g. in the case of extending the nowcasting task into short-term forecasting by predicting beyond "now".

Nowcasting with covariates is not novel, but here we propose a Bayesian hierarchical model with the advantage that it allows the direct specification of separate models for (1) the expected total case counts with reference time $t$ and (2) the time-varying delay distribution in an intuitive and well-interpretable way. The user can thus incorporate knowledge of the reporting process (weekday effects or known non-reporting days) directly in the model for reporting delay distribution. In S1 Appendix Sec 7 we derive a theoretical comparison of the nowcasting method using covariates by Bastos et al. [19]. Future work could also consist of an empirical comparison of the predictive performance of this and other nowcasting models.

Our Nowcasting method with leading indicators is flexible in terms of its application and thus can be a helpful tool for future pandemic stress situations. We support this by providing open-source software for the real-time analysis of surveillance data. Weekly updated nowcast estimates of COVID-19 fatalities and ICU admissions in Sweden using our proposed method, model RL, are found at

https://staff.math.su.se/fanny.bergstrom/covid19-nowcasting

These graphs help provide the desired situational awareness and are to be interpreted as new variants emerge.

## Supporting information

**S1 Fig. RMSE.** Average RMSE of the last 7 days; $T, \ldots, T-6$ for each reporting day $T$ in the evaluation period.
(TIF)

**S1 Appendix. Supplementary material and results.** The priors used in the Bayesian hierarchical models is found in Sec 1. In Sec 2 we show an illustration of the reporting triangle for Swedish COVID-19 deaths. Sec 3 contains detailed information about the nowcasts evaluated at day 2020-12-30 including a figure of the cumulative reporting probability and a table of the evaluation metrics, PI coverage and running times. Detailed results of the estimated regression coefficients of model L(ICU) and RL(ICU) over the evaluation period are found in Sec 4. Sec 5 covers results of including reported cases and the combination of reported cases and ICU admissions as leading indicators. In Sec 6 we show preliminary results of extending the simple random walk into a AR(2) model. Finally, a theoretical comparison of our method and the nowcasting method with covariates by Bastos et al. [19] is found in Sec 7.
(PDF)

## Acknowledgments

We thank Markus Lindroos for discussions and his contribution in coding of the reporting delay distribution.

## Author Contributions

**Conceptualization:** Fanny Bergström, Felix Günther, Michael Höhle, Tom Britton.

**Data curation:** Fanny Bergström.

**Formal analysis:** Fanny Bergström.

**Investigation:** Fanny Bergström.

**Methodology:** Fanny Bergström, Felix Günther, Michael Höhle, Tom Britton.

**Project administration:** Fanny Bergström.

**Software:** Fanny Bergström, Felix Günther, Michael Höhle.

**Supervision:** Michael Höhle, Tom Britton.

**Validation:** Fanny Bergström, Michael Höhle.

**Visualization:** Fanny Bergström, Felix Günther.

**Writing – original draft:** Fanny Bergström.

**Writing – review & editing:** Fanny Bergström, Felix Günther, Michael Höhle, Tom Britton.

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
