## [Decision Letter · Decision Letter 0]

11 Sep 2022

Dear Bergström,

Thank you very much for submitting your manuscript "Nowcasting with leading indicators applied to COVID-19 fatalities in Sweden" for consideration at PLOS Computational Biology.

As with all papers reviewed by the journal, your manuscript was reviewed by members of the editorial board and by several independent reviewers. In light of the reviews (below this email), we would like to invite the resubmission of a significantly-revised version that takes into account the reviewers' comments.

We cannot make any decision about publication until we have seen the revised manuscript and your response to the reviewers' comments. Your revised manuscript is also likely to be sent to reviewers for further evaluation.

Sincerely,

Claudio José Struchiner, M.D., Sc.D.

Academic Editor

PLOS Computational Biology

Rob De Boer

Section Editor

PLOS Computational Biology

Reviewer's Responses to Questions

**Comments to the Authors:**

Reviewer #1: I read with great interest the manuscript PCOMPBIOL-D-22-01107, "Nowcasting with leading indicators applied to COVID-19 fatalities in Sweden". The authors extended the method described in Günther et al. (2020) adding the possibility to include covariates in to the model. The inference procedure is done via MCMC and the authors have implemented their model in R-Stan. Their motivation was to provide delay corrected estimates for the daily number of deaths due to COVID-19 in Sweden. I think it is an important topic and I would like to comment some points that I think should be considered in their manuscript. The timing of the manuscript is interesting as well, since the number of cases and deaths due to COVID is increasing now in Sweden.

1) There is a very similar nowcast model based on the chain-ladder model that already takes into account covariates and also spatial random effects in the mean component. However, the aforementioned paper is pre COVID where the authors apply their method on dengue fever and on severe acute respiratory illness (SARI), Bastos et al. (2019). Miller et al. (2022) use that method to correct delays of Chikungunya fever notification in Brazil by using Google searchers and Tweets to improve the nowcasting estimates. The point here is that incorporating regression components in this class of nowcasting models is not the main novelty here, but having said that the use of such methods to improve estimates of COVID-19 fatalities is very important and worth exploring.

2) The authors should explain more how the delay was calculated. Since there are some days as I understood which new datasets are not provided (weekends and bank holidays) So there may be some "holes" in the matrix described in Fig. 3. For some lines there would not have values for certain columns. For example, if day t<t is="" monday="">

3) The author present three models, model R where log(lambda_t) follows a first order random walk, model L(m_i) where log(lambda_t) doesn't depend directly on the past but there are k-leading covariates and model RL(m_i) combining both. Is the computation time similar among them? Of course that would depend on the dimension of (m_i).

4) Is this model fast? In Bastos et al. (2019) R-INLA was used because an MCMC approach would be too timely consuming and that wouldn't be efficient on a large surveillance system (a MCMC approach was implemented on NIMBLE and the computational cost of the two approaches was very clear). In this manuscript the authors have implemented their approach in RStan which is good idea since Stan is faster than other MCMC softwares and require less iterations due to the implemented Hamiltonian Monte Carlo with the No-U-turn sampler (NUTS).

5) I though that providing an website with the most up-to-date results was quite clever, specially now with an increase of cases and deaths in Sweden. However I couldn't access the code on github page indicated in the manuscript (https://github.com/fannybergstrom/nowcasting_covid19), I believe the repository is still private.

6) An overall comparison between Bastos et al. approach and the proposed approach would be interesting, but in my humble opinion not really required for this paper. Comparing all different available nowcasting methods for deaths due to COVID-19 would be a very interesting paper, but I believe it is beyond the scopus of this manuscript that focus on COVID fatalities in Sweden.

7) A description of the scoring rules used for retrospective evaluation of the nowcasts should be presented in the Materials and Methods section. Quoting Bracher et al. (2021) "Both the logS and the CRPS cannot be evaluated directly if forecasts are provided in an interval format." perhaps the authors should consider scoring rules that take into account intervals. Comparing the interval coverage may be not enough to represent the uncertainty, since my guess is that by adding a covariate uncertainty of the nowcast estimates would be reduced, i.e. the size of the intervals would be smaller, and that would make a difference since according to the criteria presented in Table 1 the models seem to behave quite similar, Fig 7 suggests that models L(ICU) and RL(ICU) in general perform better than R model, but it is difficult to decide either RL or L model is better, a measure that quantifies the uncertainty could point out which one stands out.

8) As the authors mentioned the ICU admissions also suffer from delay. The proposal approach seems good since if we take the natural history of the disease there a time between ICU admission until the death due to COVID, so the ICU delay may be ignored. However, a two-step process could be consider where the R model would be run to ICU data, and then the corrected estimates (ICU*) would be used in models L(ICU*) and RL(ICU*). I believe this joint approach could easily be coded in RStan.

9) In equation (1) and (3) a first order random walk is assumed for log(lambda_t), I wonder if a second order random walk would bring smoother estimates and then improve the estimates.

10) Priors. What prior distributions were used for sigma (random effects variance), beta's (regression coefficients in models L and RL), phi (negative binomial overdispersion parameter) and gamma_d (equation 4). I am assuming the eta parameters (equation 4) were not used in the COVID fatality models right?

References:

Bastos et al (2019) https://doi.org/10.1002/sim.8303

Bracher et al. (2021) https://doi.org/10.1371/journal.pcbi.1008618

Miller et al. (2022) https://doi.org/10.1371/journal.pntd.0010441</t>

Reviewer #2: As usual, since the identity of the authors is known to me, I will be signing this review in the interest of fairness.

Best,

Luiz Max Carvalho.

### Major comments

In this a well-written paper, Bergstrom and colleagues address the issue of nowcasting COVID-19 in Sweden using a flexible modelling strategy that includes information on ICU admission to produce better nowcasts of case numbers.

While I commend the authors for their clear presentation and well-made figures, I would like to point out that the methodology developed on pages 5 and 6 can be considered a special case of the methodology put forth by Bastos et al. (2019, Statistics in Medicine). The omission of this citation is in my opinion a major oversight that needs immediate addressing.

Moreover, since methodologically the paper does not add anything new to the state-of-the-art, its merits must lie with its empirical findings.

On that front, I am uncertain as to what exactly is the advantage of RL(ICU) compared to R. I suppose it doesn't hurt to include ICU information, as long this is done carefully -- look at the performance of L(ICU).

In summary, I regard this as a well-written paper that unfortunately fails to mention a crucial piece of literature and therefore misses the opportunity to improve on the state-of-the-art.

### Minor comments

- These models can be implemented in INLA (https://www.r-inla.org/) which is much faster than Stan. I appreciate the Stan implementation (i) allows for more complex models to be implemented if desired and (ii) is (probably) plenty fast already. And that is why this is listed as a minor point;

- I really like the use of CRPS for (retrospectively) assessing model predictions. The fact that it is a proper scoring rule should be emphasised more, I think;

- The repository the authors point to for the code does not exist;

- I have marked up a few English mistakes/typos/awkward uses. See attached PDF.

**References**

Bastos, L. S., Economou, T., Gomes, M. F., Villela, D. A., Coelho, F. C., Cruz, O. G., ... & Codeço, C. T. (2019). A modelling approach for correcting reporting delays in disease surveillance data. Statistics in Medicine, 38(22), 4363-4377.

Reviewer #3: The authors present a nice expansion of the Nowcasting method introduced by Gunther et al. in the Biometrical Journal in 2021 and apply it to a new data set from Sweden. The introduction effectively motivates the need for generating plausible estimates of the current levels of mortality and ICU admittance given delays in reporting. Figure 1 makes the reporting lag issue very clear. But there are many COVID-19 Nowcasting papers -- a quick PubMed search returns 66 results. So what is novel here? Primarily, the authors incorporate leading indicators as covariates and compare performance to the Gunther et al. model and a hybrid of the two. The Gunther et al. model is highly cited, and improvements on this methodology could contribute to better results in the literature moving forward. The statistics used to evaluate model performance are nicely chosen and presented, indicating a slight improvement by using the hybrid approach.

My primary challenge in evaluating the updated methodology is understanding Equations 3 and 4 and the number of parameters being estimated in each of the models. Plots in the Supplement indicate time-varying coefficients while the equations do not indicate variation in the coefficient values with a time index. Unfortunately, it appears the repo with the code is currently private so I was unable to evaluate alignment between the described methodology and the actual implementation. I would request that the authors make the repo accessible and allow for reassessment of the new methodology, as the model specification is not entirely clear from the written description. Aside from the need to more closely interrogate the novel model, I only have minor revisions for the authors and believe that with a clearly understanding of the core equations I will enthusiastically recommend acceptance.

Minor revisions:

Line 17: "Nowcasting methods _have_ been used"

Line 18: No comma

Line 22-23: Revise for clarity

Fig 2: Recommend scaling to max value in first peak to see the relationship between all three before vaccines are introduced

Fig 3: Red box is looking orange

Equations 2 and 3: I would use the matrix notation with a capital M to align with equation 4 and shift the model naming convention to be L(M) and RL(M)

Equation 4: Is W a vector or a matrix?

Line 169: Consider using a percent of the total instead of a count

Line 186: "_are_shown in Fig 4"

**Have the authors made all data and (if applicable) computational code underlying the findings in their manuscript fully available?**

Reviewer #1: **No: **They mentioned in the manuscript that data and code are on github but the github page provided in the manuscript is not working.

Reviewer #2: **No: **The github link is dead.

Reviewer #3: **No: **It appears the hyperlinked repository is currently private

PLOS authors have the option to publish the peer review history of their article (what does this mean?). If published, this will include your full peer review and any attached files.

Reviewer #1: **Yes: **Leonardo S Bastos

Reviewer #2: **Yes: **Luiz Max Carvalho

Reviewer #3: No
---

## [Decision Letter · Decision Letter 1]

28 Nov 2022

Dear Bergström,

We are pleased to inform you that your manuscript 'Bayesian nowcasting with leading indicators applied to COVID-19 fatalities in Sweden' has been provisionally accepted for publication in PLOS Computational Biology.

Best regards,

Claudio José Struchiner, M.D., Sc.D.

Academic Editor

PLOS Computational Biology

Rob De Boer

Section Editor

PLOS Computational Biology

Reviewer's Responses to Questions

**Comments to the Authors:**

Reviewer #2: I'm satisfied with the modifications provided by the authors.

Reviewer #3: All comments were addressed. Thank you!

**Have the authors made all data and (if applicable) computational code underlying the findings in their manuscript fully available?**

Reviewer #2: Yes

Reviewer #3: Yes

PLOS authors have the option to publish the peer review history of their article (what does this mean?). If published, this will include your full peer review and any attached files.

Reviewer #2: **Yes: **Luiz Max Carvalho

Reviewer #3: **Yes: **Austin Carter

---

## [Editor Report · Acceptance letter]

4 Dec 2022

PCOMPBIOL-D-22-01107R1 

Bayesian nowcasting with leading indicators applied to COVID-19 fatalities in Sweden

Dear Dr Bergström,

I am pleased to inform you that your manuscript has been formally accepted for publication in PLOS Computational Biology. Your manuscript is now with our production department and you will be notified of the publication date in due course.

With kind regards,

Zsofia Freund
